# Radiological Features of Male Breast Neoplasms: How to Improve the Management of a Rare Disease

**DOI:** 10.3390/diagnostics14010104

**Published:** 2024-01-03

**Authors:** Luca Nicosia, Luciano Mariano, Anna Carla Bozzini, Filippo Pesapane, Vincenzo Bagnardi, Samuele Frassoni, Chiara Oriecuia, Valeria Dominelli, Antuono Latronico, Simone Palma, Massimo Venturini, Federico Fontana, Francesca Priolo, Ida Abiuso, Claudia Sangalli, Enrico Cassano

**Affiliations:** 1Breast Imaging Division, Radiology Department, IEO European Institute of Oncology IRCCS, 20141 Milan, Italy; luciano.mariano@ieo.it (L.M.); anna.bozzini@ieo.it (A.C.B.); filippo.pesapane@ieo.it (F.P.); valeria.dominelli@ieo.it (V.D.); antuono.latronico@ieo.it (A.L.); francesca.priolo@ieo.it (F.P.); enrico.cassano@ieo.it (E.C.); 2Department of Statistics and Quantitative Methods, University of Milan-Bicocca, 20126 Milan, Italy; vincenzo.bagnardi@ieo.it (V.B.); samuele.frassoni@unimib.it (S.F.); 3Department of Clinical and Experimental Sciences, University of Brescia, 25121 Brescia, Italy; chiara.oriecuia@unibs.it; 4Department of Molecular and Translational Medicine, University of Brescia, 25121 Brescia, Italy; 5Department of Bioimaging, Radiation Oncology and Hematology, UOC of Radiologia, Fondazione Policlinico Universitario A. Gemelli IRCSS, Largo A. Gemelli 8, 00168 Rome, Italy; simone.palma@guest.policlinicogemelli.it; 6Department of Diagnostic and Interventional Radiology, University of Insubria, Ospedale di Circolo e Fondazione Macchi, 21100 Varese, Italy; massimo.venturini@uninsubria.it (M.V.); federico.fontana@uninsubria.it (F.F.); 7Radiology Department, Università degli Studi di Torino, 10129 Turin, Italy; ida.abiuso@ieo.it; 8Data Management, European Institute of Oncology IRCCS, 20141 Milan, Italy; claudia.sangalli@ieo.it

**Keywords:** male breast cancer, breast ultrasound, mammography

## Abstract

The primary aim of our study was to assess the main mammographic and ultrasonographic features of invasive male breast malignancies. The secondary aim was to evaluate whether a specific radiological presentation would be associated with a worse receptor profile. Radiological images (mammography and/or ultrasound) of all patients who underwent surgery for male invasive breast cancer in our institution between 2008 and 2023 were retrospectively analyzed by two breast radiologists in consensus. All significant features of radiological presentation known in the literature were re-evaluated. Fifty-six patients were selected. The mean age at surgery of patients was 69 years (range: 35–81); in 82% of cases (46 patients), the histologic outcome was invasive ductal carcinoma. A total of 28 out of 56 (50%) patients had preoperative mammography; in 9/28 cases (32%), we found a mass with microcalcifications on mammography. The mass presented high density in 25 out of 28 patients (89%); the mass showed irregular margins in 15/28 (54%) cases. A total of 46 out of 56 patients had preoperative ultrasounds. The lesion showed a solid mass in 41/46 (89%) cases. In 5/46 patients (11%), the lesion was a mass with a mixed (partly liquid–partly solid) structure. We did not find any statistically significant correlation between major types of radiological presentation and tumor receptor arrangement. Knowledge of the main radiologic presentation patterns of malignant male breast neoplasm can help better manage this type of disease, which is rare but whose incidence is increasing.

## 1. Introduction

Worldwide, there are more than two million new diagnoses of female malignancies annually, with more than 600,000 deaths [1]. In contrast, male breast neoplasia is rare, with mechanisms of presentation and management mode not entirely clear or globally agreed upon. Less than 1% of all breast cancer (BC) patients are male, and BC represents only about 0.3% of all cancers in men, with a prevalence in Europe of 1 in 100,000 cases [2]. The prevalence of male breast neoplasia appears to be slightly higher in some African populations, such as Zambia, and in the Jewish population [3]. By contrast, it seems to be somewhat lower in Japan [4].

Globally, however, especially in Western populations, there has been an upward trend in the incidence of this neoplasm in recent years, in tandem with the increase in life expectancy [5].

Unfortunately, breast neoplasms most often present with symptoms, resulting in a possible worsening in prognosis and finding lymph node metastasis already at initial diagnosis. This delay can lead to disease progression that can result in a worse prognosis [6].

Therefore, the scientific community’s effort should be to spread the culture of prevention, especially in high-risk patients [7]. According to the literature, patients with the most significant risk of developing this type of neoplasm are those with a BRCA mutation, Klinefelter’s syndrome, testicular and liver disease, and previous chest radiotherapy. Chronic alcohol intake, obesity, and exogenous estrogen intake are lifestyle-related risk factors [2].

Given the rarity of this condition, physicians dedicated to the management of this condition often need to be made aware of the main modes of its presentation [8]: neoplastic pathology can often be confused with more common benign conditions such as gynecomastia [8].

This paper aims to present our single-center case experience in a tertiary referral cancer hospital of the main radiologic presentation patterns of malignant invasive male breast neoplasms.

Furthermore, our objective is to contribute valuable insights to the existing body of knowledge by analyzing whether distinct radiological presentation patterns in malignant invasive male breast neoplasms are linked to unfavorable receptor patterning, thereby enriching the understanding of the disease’s molecular characteristics and potential implications for prognosis and treatment strategies.

Based on our results, we would like to propose a possible prevention protocol for patients at higher risk of developing breast malignancy. We want to provide physicians involved in managing this type of disease with the tools to promptly recognize the radiological presentation of the most aggressive forms of male malignancies.

## 2. Materials and Methods

This retrospective study was conducted according to the guidelines of the Declaration of Helsinki and approved by the local Ethics Committee (approval number: UID 3905).

All patients undergoing surgery for infiltrating male BC in our center between 2008 and 2023 were selected for retrospective analysis of radiological images.

The Digital Mammography (DM) unit used for these analyses was the GE^®^ Healthcare, Senographe Pristina^®^ (Chalfont St. Giles, UK). The DM image acquisition protocol involved a cranio-caudal and medio-lateral oblique projection on each side. Considering the reduced thickness of the male breast tissue, the median exposure was 30 kPv (Kilovoltage peak) and 90 mAS (milliamperes/second).

The ultrasound (US) exams were performed using an Esaote with MyLab™X7 (Esaote^®^, Genova, Italy). A high-frequency (10 MHz to 15 MHz) linear probe (LMX 4-20 XCrystal) was used for US evaluation.

The images were retrospectively analyzed according to the conventional BI-RADS [9,10] by two breast radiologists in consensus (A.B. with 25 years of experience in breast imaging and L.N. with 8 years of experience).

Patients with insufficient radiological images for retrospective image re-evaluation or whose images were not stored in the institution’s picture archiving and communication (PACS) were excluded. Patients with male BC in situ were also excluded because they were the subject of our previous publication [11].

The following parameters analyzed DM lesions:-Type of lesion (mass, mass with microcalcifications);-Morphology of microcalcifications if present (amorphous, round and punctate);-Distribution of microcalcifications (clustered, scattered);-Relationship of the lesion to the nipple (eccentric, subareolar). The lesion was defined as eccentric if assuming an imaginary line from the center of the nipple that does not fall within the lesion;-Shape of the lesion (oval, round, spiculated);-Density lesion (high, low);-Margins (regular, poorly defined, irregular);-Associated findings (skin thickening, nipple involvement).

US lesions were analyzed using the following parameters:-Type of lesion (mixed, partly liquid–partly solid nodule, solid nodule);-Margins (spiculated, smooth, poorly defined);-Lesion taller than wide (yes, no);-Echogenicity (homogeneous, inhomogeneous);-Color Doppler (absent, predominantly in the rim, internal);-Posterior acoustic enhancement (yes, no).

Other variables were collected: patients’ risk conditions (where available, given the retrospective nature of the analysis of clinical records) such as familiarity (at least one first-degree relative with breast neoplasm), BRCA mutations, and testicular disease; age at surgery; and data regarding type and year of surgery, lymph node disease involvement at surgery, histologic type of neoplasm with G grading system associated (G1: low grade; G2: intermediate grade; G3: high grade), receptor patterns, follow-up information to calculate the disease-free survival (DFS) (we monitored the eventual occurrence of homo- and contralateral locoregional disease recurrence, single-site or multiple-site metastasis, and death). Given the low number of deaths, we did not calculate overall survival (OS).

For our analysis, we considered patients with Ki-67 > 20% [12] and with G3 grading [13] as patients with the worst prognosis. The surgical specimen was considered the gold standard for histological analysis and the evaluation of the receptor status.

## 3. Results

The radiological images of 56 patients were analyzed. Figure 1 shows the flowchart diagram with the inclusion/exclusion criteria of the study.

### 3.1. Overall Summary (Data Not Related to Imaging)

The mean age at surgery of patients was 69 years (range: 35–81); 28 patients (50% of cases) had preoperative DM; and 46 patients (82% of cases) had preoperative US. The median lesion size was 18 mm (range: 5–70). All patients presented with a clinically palpable lesion. A preoperative cytologic assessment was performed in most cases (59% of cases), and less frequently, a preoperative core biopsy was performed (41%). Most patients (96% of cases) underwent a mastectomy.

The histologic outcome was invasive ductal carcinoma in 82% of cases, and the second most frequent histotype was invasive papillary carcinoma (13%). Lymph node metastasis at surgery was found in 32% of cases.

Even considering a large number of missing data due to the study’s retrospective nature, in 53% (25/47) of the cases, patients had a family history of breast malignancy. Patients in 33% (6/18) of the cases had a BRCA mutation. Prostate hyperplasia was found in 44% (8/18) of cases.

The most frequent receptor pattern was ER/PgR-positive (93%) and HER2-negative (96%) [14]. In 29% of cases, patients presented with grade G3 neoplasm, and in 64% of cases with grade G2 neoplasm. In 54% of cases, Ki-67 was greater than or equal to 20%.

The general descriptive data of our population are summarized in Table 1.

### 3.2. DM Features

Considering only the 28 patients with preoperative DM, the lesion presented as a mass in all cases. In 9/28 cases (32%), we found a mass with microcalcifications on DM. In 78% (7/9) of the masses with microcalcifications, the microcalcifications presented as round and punctate, and in 22% (2/9) cases, as amorphous. The microcalcifications presented as scattered distributions in 7/9 (78%) of the masses with microlesions. In 18 cases (64%), the relationship of the mass to the nipple was eccentric. In 18 cases (64%), the shape of the mass was spiculated. In 89% of cases, the mass presented high density, and in 15/28 (54%) cases, the mass gave irregular margins. Finally, there were associated findings in 8/28 (29%) cases, such as skin thickening or nipple involvement. We summarize the DM presentation patterns of male neoplasms in Table 2. In 61% of cases, the lesion was classified as BI-RADS 4c.

### 3.3. US Features

Considering only the 46 patients with preoperative US, the lesion presented as a solid mass in 41/46 (89%) cases. In 11% of cases, it presented as a mass with a mixed (partly liquid–partly solid) structure. In 96% of cases, the lesion presented inhomogeneous echogenicity. In 31/46 (67%) cases, the lesion presented with ill-defined or spiculated margins. The lesion was more expansive than tall in 29 of 46 cases (63%). In 40 cases, the lesion was shown to be vascularized on Color Doppler; in 26 cases, vascularization was mainly located in the rim (60%), while in 14 cases, the vascularization was predominantly internal (33%). Finally, posterior acoustic enhancement was found in 5 of 46 cases (11%). We summarized the US presentation patterns of male neoplasms in Table 3.

We analyzed the relationship between radiological aspects of the neoplasm and tumor receptor profile.

We found no statistically significant correlation between major types of radiologic presentation and tumor receptor patterns (Table 4).

For example, among patients who have a relation of the mass to the nipple = “Eccentric”, 28% have G3, and among patients with the relation “subareolar”, 50% have G3 (*p*-value 0.41). Among patients who have a mass with microcalcifications, 44% have Ki-67 > 20%, while among patients with mass without microcalcifications, 44% have Ki-67 > 20%. (*p* = 1).

### 3.4. Disease-Free Survival

Nine patients (16%) had a DFS event during a median follow-up of 4 years (interquartile range: 1.1–7.0): one loco-regional lymph node metastasis, two bone metastases, one liver metastasis, one lung metastasis, one multiple metastasis, and three deaths as a first event.

The 1-, 3-, 8-, and 10-year disease-free survivals were, respectively, 96% (85–99), 89% (75–95), 81% (62–91), and 73% (48–87) (see Figure 2).

## 4. Discussion

Male breast neoplasms are rare, yet their incidence is increasing due to the increase in the population’s average life expectancy [5].

According to some statistics, annual deaths from male BC could be comparable to those from testicular neoplasms [15].

In almost all cases, patients present late to medical attention when there are already obvious clinical symptoms of breast neoplasm [16]. This often leads to a non-early-stage disease diagnosis, with overt lymph node disease associated [17,18]. In our study, node metastasis was present in more than 30% of patients.

Thus, the effort of the scientific community should be to promote male breast neoplasia prevention with appropriate radiological examinations, especially for high-risk patients (e.g., patients with BRCA mutation, Klinefelter’s syndrome, testicular and liver disease, patients who have previously undergone chest radiotherapy, patients who consume large amounts of alcohol, obese patients, and patients taking estrogen). Currently, no recommendations guide male breast screening in asymptomatic high-risk patients. However, preliminary studies have shown that preventive DM could be of great benefit in providing early diagnosis and, thus, a better prognosis [19].

With our study, we wanted to provide an overview of the main radiological features of infiltrating male breast neoplasms; knowledge of these feature presentations may allow us to better cope with malignant male pathology and to have tools to distinguish it from benign lesions. To the best of our knowledge, this is one of the works with the most significant number of patients with male invasive breast neoplasm studied in a single center.

The results show that some of the most common modes of presentation of male breast neoplasms are similar to those of females (e.g., high density, irregular margins, skin thickening, nipple retraction) [20,21].

In our experience, 96% of lesions are inhomogeneous; in more than 90% of cases, they are vascularized on Color Doppler, with high density on DM (89% of cases) and spiculated margins (64% of cases). However, in non-negligible percentages in our study, the malignant lesion presented typical benign female lesion characteristics: margins proved to be regular in 32% and smooth in 33% of cases; in many cases (63%), the mass proved to be wider than taller and in 11% of cases the lesion presented with a cystic component. The associated microcalcifications were often scattered and punctate (a typical benign sign of female microcalcifications). Male BC rarely presents as a mass with microcalcifications associated or with only microcalcifications.

Male breast carcinoma does not infrequently have a cystic component visible on US. The well-defined margins of the lesion should also not be misleading, as they are often associated with infiltrating male breast neoplasms.

Typical and atypical BC presentations are shown in Figure 3, Figure 4, Figure 5, Figure 6, Figure 7 and Figure 8.

The figure shows medio-lateral oblique (a) and cranio-caudal projections of the right breast (b). A mass with high density, eccentric to the nipple with spiculated margins and without microcalcifications (arrow), is presented. We can also identify a pathological lymph node (arrowhead) in the right axillary region. This is a typical presentation of male infiltrative BC DM. At surgery, an infiltrative ductal carcinoma was diagnosed (receptor arrangement: ER 95%; PgR 90%; Ki-67 23%; Her 2 negative; grading G2) with positive histological evaluation of the axillary lymph node for BC metastases.

The figure shows medio-lateral oblique (a) and cranio-caudal (b) projections of the right breast. A round mass (arrow) with regular and well-defined margins (features most readily associated with female benign lesions) in the central subareolar area (a location where gynecomastia is most frequently appreciated) is presented. This is a less common type of male BC DM presentation. At surgery, an infiltrating ductal carcinoma was diagnosed (receptor arrangement: ER: 90%; PgR: 90%; Ki-67: 25%; Her 2 negative; grading G3).

The figure shows a cranio-caudal projection of the right breast with an eccentric, high-density mass (arrow) and ill-defined margins. Scattered and partly clustered amorphous microcalcifications (arrowheads) are located within the mass. The association of microcalcifications in male BC is rare. At surgery, an infiltrative ductal carcinoma was diagnosed (receptor arrangement: ER: 95%; PgR 30%; Her 2 negative; Ki-67%: 40%; grading G3).

Figure 6a shows a medio-lateral oblique projection of the right breast with a subareolar opacity (arrow), non-defined margins, and low/intermediate density (typical presentation pattern of male gynecomastia). Figure 6b shows a medio-lateral oblique projection of the left breast with retroareolar eccentric opacity, spiculated margins, and high density. Skin thickening and retraction (arrowhead) were associated (typical presentation of male BC). At surgery, an infiltrative ductal carcinoma was diagnosed (receptor pattern ER 95%; PgR 95%; Ki-67: 25%; Her 2 negative; grading G3).

The US image shows a superficial inhomogeneous nodule with a solid component (arrowhead), liquid component (arrow), and well-defined margins. The posterior acoustic enhancement given by the liquid component can also be seen (asterisk). At surgery, an infiltrative papillary carcinoma was diagnosed (receptor arrangement: ER 95%; PgR: 95%; Ki-67 7%; Her 2 negative; grading G1).

The US image shows a solid nodule with irregular margins and predominant rim vascularization on Color Doppler. At surgery, an infiltrative ductal carcinoma was diagnosed (receptor arrangement: ER: 80%; PgR: 60%; Ki-67: 35%; Her 2 negative; grading G2).

Therefore, whenever a male breast lesion is found even with well-defined margins, cystic components, or diffuse, scattered, and punctate microcalcifications (typical findings of female benign neoplasm), it should be further investigated.

The findings presented are consistent with the limited body of research available in the current literature. In a study by Mathew et al. [22] focusing on 51 instances of male breast neoplasms, the shape and margins of the masses were identified as oval and circumscribed in 35% and 11% of cases, respectively. Notably, a cystic component was observed in 22% of the examined masses [22]. Yang et al., in a study involving eight patients, reported a cystic mass in 50% of cases [23]. Lastly, in the investigation by Sahin et al. [24], mass opacities with microcalcifications were discerned in only 1/25 patients (4%).

Interestingly, none of the radiological features of presentations of our case history are associated with a good receptor profile, even those hypothetically less severe, such as smooth margins and cystic components (Table 4); therefore, radiological presentations linked to more typical benign features may also be connected with an aggressive receptor profile (HER2-positive or triple-negative patterns).

The patient’s age (over 60 years), the presence of a mass lesion, often eccentric to the nipple position, with high DM density, nipple retraction and skin thickening, are the main findings that allow us to distinguish a malignant from a benign male lesion (e.g., gynecomastia) [25].

Based on our results, we can conclude that both DM and US provide comprehensive information for the analysis of male breast lesions and should also be considered preventive examinations in asymptomatic high-risk patients [26]. The benefits of screening DM (especially the high sensitivity for malignancy detection, 94.7%, NPV of 99.7%) outweigh the potential drawbacks of false positives, costs, and radiation exposure [7,26]. Furthermore, considering the infrequency of an imaging manifestation featuring microcalcifications, US could be a crucial adjunct to DM in systematic preventive screening programs for asymptomatic patients [27]. Its better specificity (95.3%) for malignancy detection and simultaneous axillary lymph node status could avoid unnecessary biopsy and surgical procedures [28].

The American College of Radiology recently recommended DM or Digital Breast Tomosynthesis (DBT) in symptomatic men aged 25 and older or with a suspicious physical examination for BC [29]. DBT enhances the detection of subtle or occult findings compared to 2D imaging, such as architectural distortions, thereby minimizing the superimposition of fibroglandular tissue, especially in dense breasts or with superimposed gynecomastia [30,31]. Incorporating DBT into conventional 2D DM lowers recall rates, improving BC detection [28,32]. Tari et al. proposed a single simple algorithm in both symptomatic and asymptomatic patients at high risk who are older than 25 years old, suggesting performing a physical examination with at least a single MLO projection for both breasts, better if in DBT [28]. US or other DM views should be performed secondarily.

DM and US utilize subjective criteria defined by BI-RADS to achieve consistent interpretation outcomes and distinguish and stratify the risk of related abnormalities. Ultrasound tomography (UST) is an emerging technique that moves beyond B-mode imaging by its transmission capabilities, providing potential for tissue-specific imaging and characterization [33]. Conventional reflection US provides anatomical images of breast tumors based on differences in acoustic impedance among tissues [34,35]. Utilizing transmission to measure parameters such as sound velocity and attenuation [36,37], UST allows for detailed characterization, enabling precise differentiation between cancerous tissue and normal or benign conditions. Studies in the literature demonstrated a high degree of correlation of breast tissue structures compared to fat-subtracted contrast-enhanced MRI, with a scan of ~90% of breast volume [38]. A supplementary application of UST could enhance specificity compared to existing US methods, offering a thorough screening approach that identifies invasive BC not detectable through DM. Identifying these early-stage invasive BCs could offer women the chance for curative treatment that might otherwise be missed.

Lastly, considering our patients’ DFS with a median follow-up time of 4 years, we found 73% over 10 years. This outcome surpasses those previously reported in the current literature [16], probably due to our study’s short follow-up observation time.

The main limitation of our study is its monocentric and retrospective nature: a lot of data regarding patients’ risk factors must be obtained, or a longer observation follow-up needs to be performed. Although the case series is significant for a study of this type on male breast neoplasms, the small number of patients resulted in low statistical power, i.e., a lower chance of a given clinical difference being statistically significant. Prospective, larger scale studies will be needed in the future to confirm our results in diagnostic and screening settings.

## 5. Conclusions

Male breast neoplasms, while rare, are on the rise, emphasizing the need for increased awareness and preventive measures. Our study highlights the challenges associated with late-stage diagnoses, emphasizing the importance of promoting radiological examinations for high-risk individuals. The diverse radiological features observed, including atypical presentations resembling benign lesions, underscore the need for thorough investigation even in cases with seemingly typical characteristics. Our findings suggest that certain radiological features, conventionally associated with benign lesions, may also be linked to aggressive receptor profiles.

Both DM and US play pivotal roles in providing comprehensive information for the analysis of male breast lesions. DBT emerges as a valuable adjunct, offering enhanced detection capabilities, particularly in cases with dense breasts or superimposed gynecomastia. UST represents a promising avenue, surpassing traditional B-mode imaging by offering tissue-specific imaging and characterization. Its potential to enhance specificity in detecting invasive BC not detectable through DM suggests a valuable role in comprehensive screening.

While our study provides insights into the radiological features of male breast neoplasms, its limitations, including its monocentric and retrospective nature and the small sample size, necessitate future prospective and larger-scale studies. These endeavors will be crucial to confirming and expanding upon our results, ultimately contributing to improved diagnostic and screening strategies for male breast neoplasms.

## Figures and Tables

**Figure 1 diagnostics-14-00104-f001:**
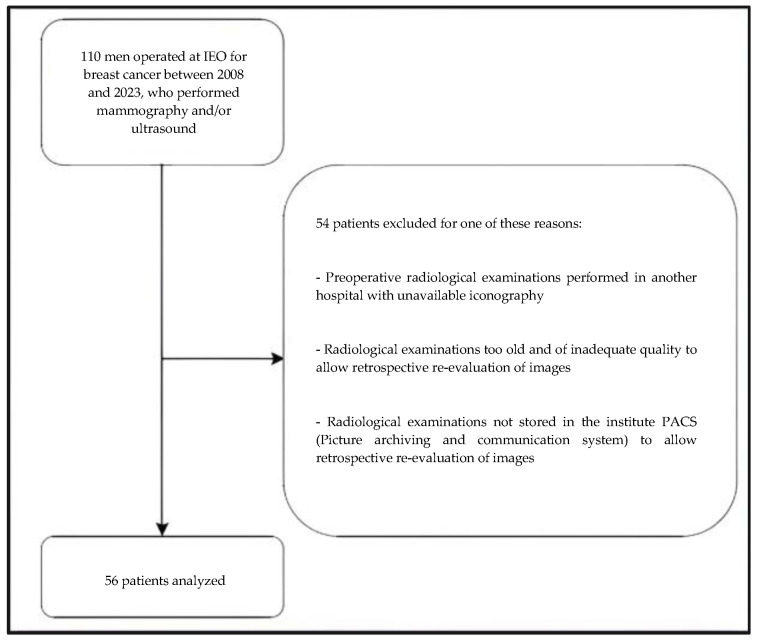
Flowchart diagram of the study inclusion and exclusion criteria.

**Figure 2 diagnostics-14-00104-f002:**
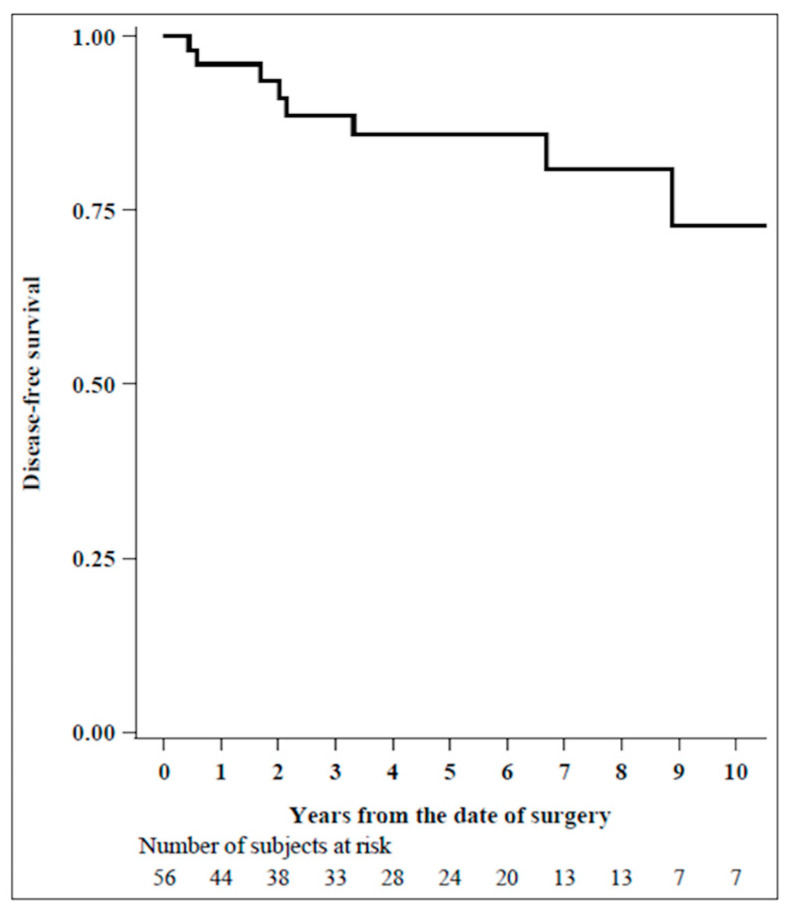
Disease-free survival (median FU (Q1–Q3) in years: 4.0 (1.1–7.0), N = 56).

**Figure 3 diagnostics-14-00104-f003:**
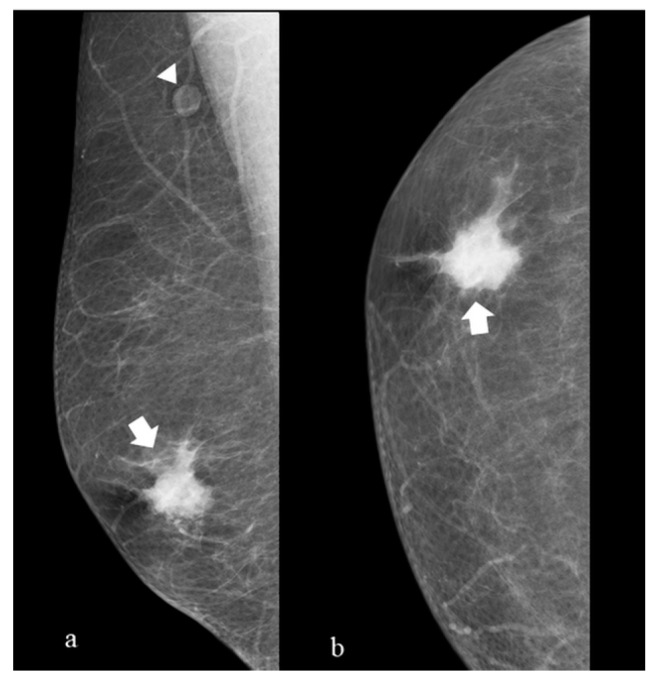
(**a**) (mediolateral oblique projection), (**b**) (cranio caudal projiection) 60-year-old man with a palpable mass of the upper sectors on the right breast (arrow).

**Figure 4 diagnostics-14-00104-f004:**
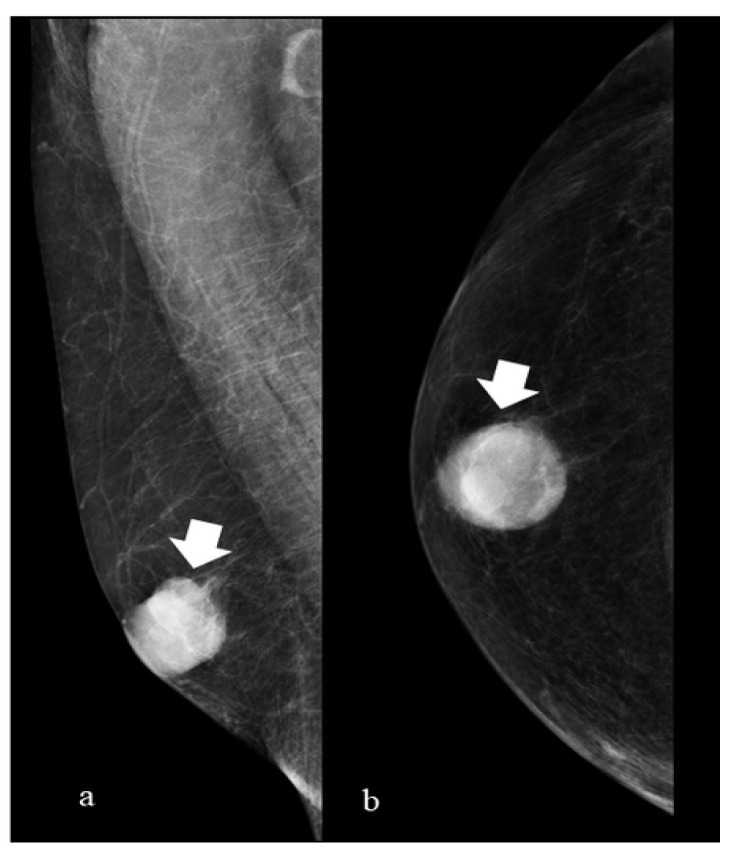
(**a**) (mediolateral oblique projection), (**b**) (cranio caudal projiection 75-year-old man with a palpable mass on the right breast (arrow).

**Figure 5 diagnostics-14-00104-f005:**
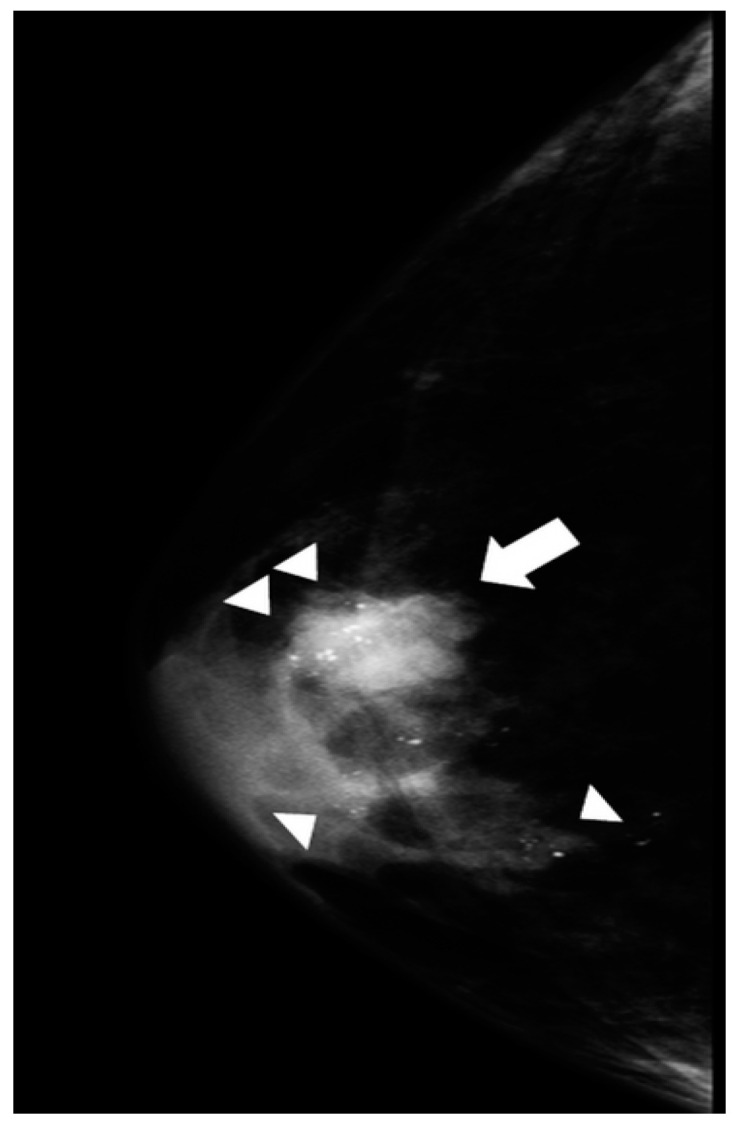
A 79-year-old man with a palpable mass on the right breast (arrow).

**Figure 6 diagnostics-14-00104-f006:**
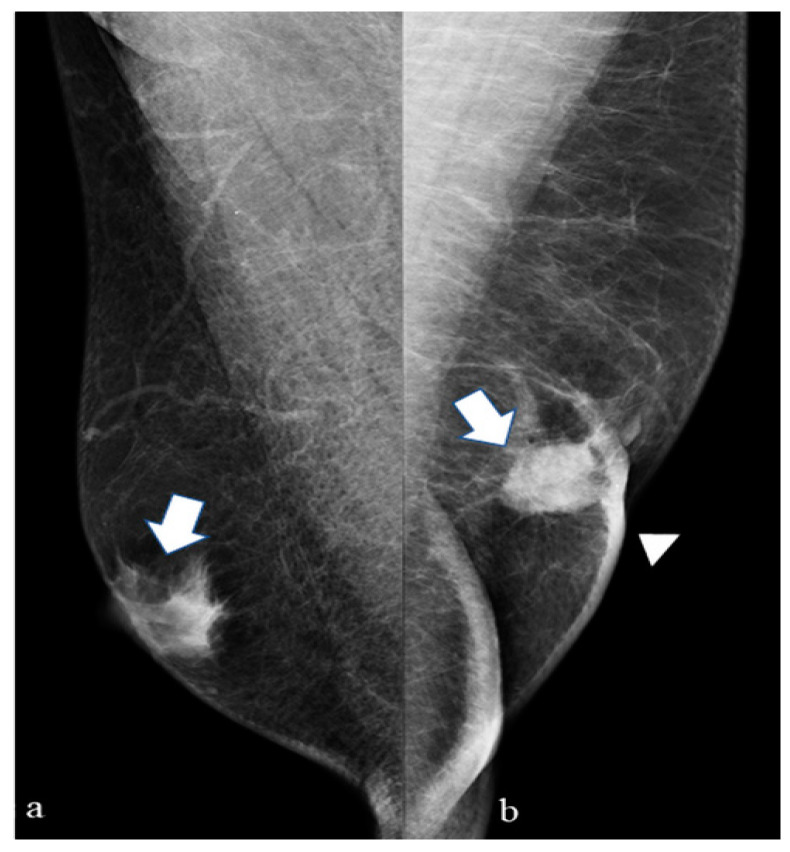
Typical differences between gynecomastia and breast neoplasm (primary differential diagnosis): (**a**) A 45-year-old patient with a retroareolar, palpable, and mobile right mass (arrow). (**b**) A 70-year-old patient with a solid mass (arrow) and left nipple retraction (arrowhead).

**Figure 7 diagnostics-14-00104-f007:**
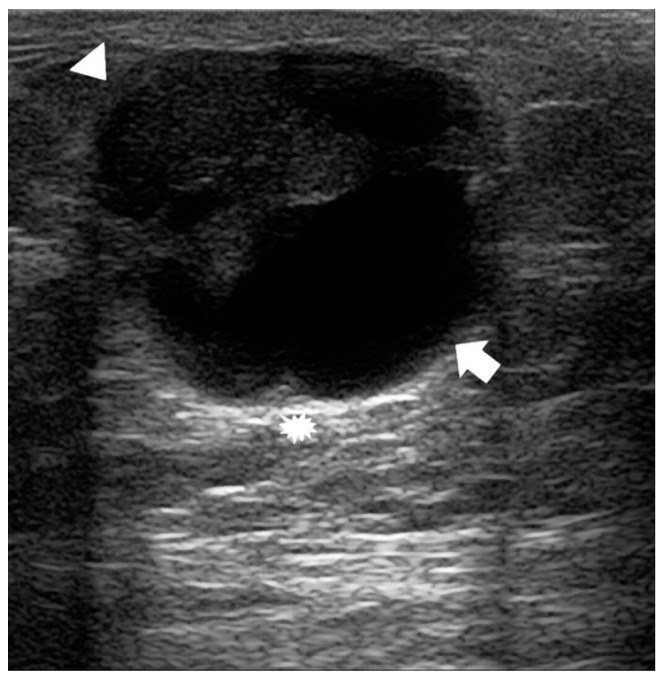
A 75-year-old man with a palpable lump and blood secretion of the right breast.

**Figure 8 diagnostics-14-00104-f008:**
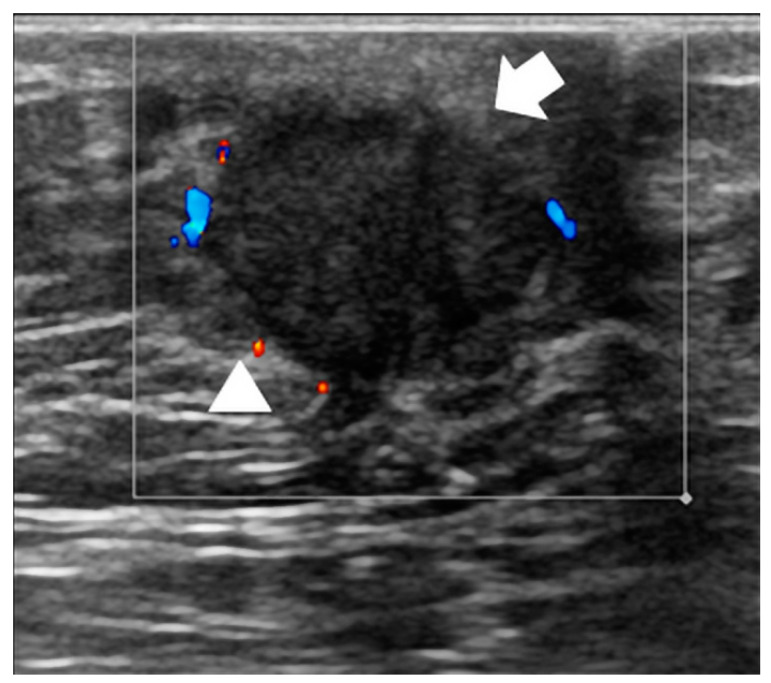
67-year-old man with a palpable mass of the right breast.

**Table 1 diagnostics-14-00104-t001:** Descriptive variables (N = 56).

Variable	Level	Overall (N = 56)
Familiarity, N (%)	No	22 (47)
	Yes	25 (53)
	Missing	9
BRCA mutations, N (%)	No	12 (67)
	Yes	6 (33)
	Missing	38
Prostatic/testicular diseases, N (%)	Prostatic hyperplasia	8 (44)
	Prostate adenoma	1 (6)
	Prostate cancer	4 (22)
	Testicular cancer	3 (17)
	Bladder cancer	1 (6)
	Varicocele	1 (6)
	Missing	38
Pre-operative mammography, N (%)	No	28 (50)
	Yes	28 (50)
Pre-operative ultrasound, N (%)	No	10 (18)
	Yes	46 (82)
Size of the lesion (mm), median (min–max)		18 (5–70)
Pre-operative assessment, N (%)	Cytology	33 (59)
	Core biopsy	23 (41)
Year of surgery, N (%)	2008–2010	9 (16)
	2011–2013	10 (18)
	2014–2016	15 (27)
	2017–2019	14 (25)
	2020–2023	8 (14)
Age at surgery (y), median (min–max)		69 (35–81)
Type of surgery, N (%)	Lumpectomy	2 (4)
	Mastectomy	54 (96)
Side, N (%)	Right	29 (52)
	Left	27 (48)
Finding of lymph node mets at surgery, N (%)	No	38 (68)
	Yes	18 (32)
Histological result, N (%)	Invasive ductal carcinoma	46 (82)
	Papillary invasive carcinoma	7 (13)
	Invasive ductal and papillary carcinoma	2 (4)
	Invasive ductal and cribriform carcinoma	1 (2)
ER (Estrogen Receptor), N (%)	<1%	0 (0)
	≥1%	56 (100)
PgR (Progesterone Receptor), N (%)	<1%	4 (7)
	≥1%	52 (93)
Ki-67, N (%)	<20%	26 (46)
	≥20%	30 (54)
HER2 status, N (%)	0/1+/2+	54 (96)
	3+	2 (4)
Grading, N (%)	G1	4 (7)
	G2	35 (64)
	G3	16 (29)
	Missing	1

**Table 2 diagnostics-14-00104-t002:** Descriptive variables among patients with pre-operative DM (N = 28).

Variable	Level	Overall (N = 28)
BI-RADS, N (%)	4a	1 (4)
	4b	5 (18)
	4c	17 (61)
	5	5 (18)
Type of mammographic lesion, N (%)	Mass	19 (68)
	Mass with microcalcifications	9 (32)
Relation of the mass to the nipple, N (%)	Eccentric	18 (64)
	Subareolar	10 (36)
Shape of the mass, N (%)	Oval	5 (18)
	Round	5 (18)
	Spiculated	18 (64)
Density of the mass, N (%)	Low	3 (11)
	High	25 (89)
Margins of the mass, N (%)	Irregular	15 (54)
	Poorly defined	4 (14)
	Regular	9 (32)
Associated findings, N (%)	No	20 (71)
	Nipple retraction	2 (7)
	Skin thickening	5 (18)
	Skin thickening and nipple involvement	1 (4)

**Table 3 diagnostics-14-00104-t003:** Descriptive variables among patients with pre-operative US (N = 46).

Variable	Level	Overall (N = 46)
BI-RADS, N (%)	4a	8 (17)
	4b	3 (7)
	4c	31 (67)
	5	4 (9)
Type of ultrasound lesion, N (%)	Solid mass	41 (89)
	Mixed mass	5 (11)
Margins of the mass, N (%)	Poorly defined	18 (39)
	Smooth	15 (33)
	Spiculated	13 (28)
Mass taller than wide, N (%)	No	29 (63)
	Yes	17 (37)
Echogenity, N (%)	Inhomogeneous	44 (96)
	Homogeneous	2 (4)
Color Doppler, N (%)	Absent	3 (7)
	Internal	14 (33)
	Predominantly in the rim	26 (60)
	Missing	3
Posterior acoustic enhancement, N (%)	No	41 (89)
	Yes	5 (11)

**Table 4 diagnostics-14-00104-t004:** Association between DM (N = 28) and US (N = 46) variables with tumor characteristics.

Variable	Level	Ki-67		*p*-Value	Grading		*p*-Value
		<20%	≥20%		G1/G2	G3	
Mammographic variables among patients with pre-operative mammography (N = 28)							
Overall, N (%)		**14 (50)**	**14 (50)**		**18 (64)**	**10 (36)**	
BI-RADS, N (%)	4a/4b/4c	12 (52)	11 (48)	1.00	15 (65)	8 (35)	1.00
	5	2 (40)	3 (60)		3 (60)	2 (40)	
Type of mammographic lesion, N (%)	Mass	9 (47)	10 (53)	1.00	13 (68)	6 (32)	0.68
	Mass with microcalcifications	5 (56)	4 (44)		5 (56)	4 (44)	
Relation of the mass to the nipple, N (%)	Eccentric	9 (50)	9 (50)	1.00	13 (72)	5 (28)	0.41
	Subareolar	5 (50)	5 (50)		5 (50)	5 (50)	
Shape of the mass, N (%)	Oval	2 (40)	3 (60)	0.76	4 (80)	1 (20)	0.53
	Round	2 (40)	3 (60)		2 (40)	3 (60)	
	Spiculated	10 (56)	8 (44)		12 (67)	6 (33)	
Margins of the mass, N (%)	Irregular	8 (53)	7 (47)	0.34	9 (60)	6 (40)	0.36
	Poorly defined	3 (75)	1 (25)		4 (100)	0 (0)	
	Regular	3 (33)	6 (67)		5 (56)	4 (44)	
Associated findings, N (%)	No	9 (45)	11 (55)	0.68	13 (65)	7 (35)	1.00
	Yes	5 (63)	3 (38)		5 (63)	3 (38)	
Ultrasound variables among patients with pre-operative ultrasound (N = 46)							
Overall, N (%)		**20 (43)**	**26 (57)**		**31 (69)**	**14 (31)**	
BI-RADS, N (%)	4a/4b/4c	18 (43)	24 (57)	1.00	29 (71)	12 (29)	0.58
	5	2 (50)	2 (50)		2 (50)	2 (50)	
Type of ultrasound lesion, N (%)	Solid mass	17 (41)	24 (59)	0.64	27 (68)	13 (33)	1.00
	Mixed mass	3 (60)	2 (40)		4 (80)	1 (20)	
Margins of the mass, N (%)	Poorly defined	8 (44)	10 (56)	0.61	12 (71)	5 (29)	0.85
	Smooth	5 (33)	10 (67)		11 (73)	4 (27)	
	Spiculated	7 (54)	6 (46)		8 (62)	5 (38)	
Mass taller than wide, N (%)	No	12 (41)	17 (59)	0.76	20 (69)	9 (31)	1.00
	Yes	8 (47)	9 (53)		11 (69)	5 (31)	
Color Doppler, N (%)	Absent	2 (67)	1 (33)	0.63	2 (100)	0 (0)	0.62
	Internal	5 (36)	9 (64)		11 (79)	3 (21)	
	Predominantly in the rim	12 (46)	14 (54)		17 (65)	9 (35)	

## Data Availability

The data presented in this study are available on request from the corresponding author. The data are not publicly available due to privacy concerns, in accordance with GDPR.

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
