# Peer review of "Radiological Features of Male Breast Neoplasms: How to Improve the Management of a Rare Disease"

_diagnostics, 2024, doi:10.3390/diagnostics14010104_

Round 1
Reviewer 1 Report
Comments and Suggestions for Authors
Please check the attached file.

Comments on the Quality of English LanguageThe overall quality of English Language is good. However, there are some errors that need to be reviewed by the authors as a whole.
Author Response
In this work, the author's goal is to investigate the features of radiographic imaging of male breast cancer to improve the likelihood of early detection and to establish prevention protocols. The results seem interesting, however, following are a few concern that should be addressed:
1.The resolution in Figure 1 is too low. Please change it to a higher resolution for reader readability.
Thank you we have changed the resolution of the figure as per your request
- If it is worth considering the following prior studies in the introduction, please check it.
- Tari et al., Male breast cancer review. A rare case of pure DCIS: imaging protocol, radiomics and management, Diagnostics, 11(12), 2021, 2199. https://doi.org/10.3390/diagnostics11122199
Thank you for the suggestion.
We have considered and discussed the article you suggested.
- Adding conditions (e.g., kVp, mAs, probe type, etc.) for generating radiographic images and ultrasound
images will help readers apply the results of the paper.
we have added the required parameters in the text.
The average exposure (considering the reduced mammary thickness of the male breast) was 30 kVp and 90 mAS. A high-frequency (10 MHz to 15 MHz) linear probe (LMX 4-20 XCrystal) was used for ultrasound examinations.
- If you write down the used post-image processing method together, it will be helpful for researchers in related fields.
Thanks, we added this part in the text.
“All acquired images were stored in the institution's PACS at the time of acquisition and subsequently re-evaluated retrospectively”.
- Please match the size of parentheses. “Table 1. Descriptive variables (N=56).”
Thanks. Done
- Please redraw the line in Table 4.
Thanks. Done
- Radiographic images and ultrasound images are known to be the most common methods for diagnosing and non-invasive breast cancer. However, breast cancer screening based on blast tomosynthesis (DBT) and ultrasound computed tomography (USCT) has also been actively studied recently. Please comment on this as well.
Thanks
We added a careful discussion on this topic in the text.
- Please carefully proofread and spell check the document to eliminate errors in the use of English. A careful read-through and correction by a native English speaker would help with readability in a few places.
Thank you.
The text was proofread by a native English speaker.
Reviewer 2 Report
Comments and Suggestions for Authors
- The study provides an overview of the main radiological features of infiltrating male breast neoplasms in order to better cope with malignant male pathology and to have tools to distinguish it from benign malignancy.
- It had been concluded that both mammography and ultrasound provide comprehensive information for the analysis of male breast lesions and should also be considered preventive examinations in asymptomatic high-risk patients.
- However, the main limitation is the small number of patients. This may lead to a lack of statistical analysis of results and may affect final decision.
- Give a full description of the images in the dataset (format, type, size, ....)
- The conclusion section needs to be extended to give a better representation about the main paper contributions.
Comments on the Quality of English LanguageThe English language is generally good. Just avoid extensive use of "we", "our" in scientific writing.
Author Response
- The study provides an overview of the main radiological features of infiltrating male breast neoplasms in order to better cope with malignant male pathology and to have tools to distinguish it from benign malignancy.
- It had been concluded that both mammography and ultrasound provide comprehensive information for the analysis of male breast lesions and should also be considered preventive examinations in asymptomatic high-risk patients.
- However, the main limitation is the small number of patients. This may lead to a lack of statistical analysis of results and may affect final decision.
Thanks for the comment. We added this part in the text.
"The small number of patients is due to the rarity of the pathology and the monocentric nature of the study. We are aware that this is a limitation of the study, and in fact we had already reported it in the discussion of the Paper. In addition, the small number of patients (although the case series is large for a study of this type on male breast neoplasms) resulted in low statistical power, i.e., a lower chance of a given clinical difference being statistically significant."
- Give a full description of the images in the dataset (format, type, size, ....)
All images were prepared in HD according to the magazine's rules, HD, 360 DPI, TIFF Format, 1,262 KB
- The conclusion section needs to be extended to give a better representation about the main paper contributions.
Thanks. We extended the conclusion section as per your suggestion.
Round 2
Reviewer 1 Report
Comments and Suggestions for Authors
This revision is well conducted according to the reviews and the requested research contents for the detail were sufficiently reflected. I think that this article deserved the consideration of published.
Author Response
We thank you for the comment which gratifies our work. As per your indication in the review panel, we have slightly modified some lines of the introduction to make it clearer and more complete.